# Differences in Unfavorable Lifestyle Changes during the COVID-19 Pandemic between People with and without Disabilities in Finland: Psychological Distress as a Mediator

**DOI:** 10.3390/ijerph19126971

**Published:** 2022-06-07

**Authors:** Marja Eliisa Holm, Päivi Sainio, Jaana Suvisaari, Katri Sääksjärvi, Tuija Jääskeläinen, Suvi Parikka, Seppo Koskinen

**Affiliations:** Department of Public Health and Welfare, Finnish Institute for Health and Welfare (THL), 00271 Helsinki, Finland; paivi.sainio@thl.fi (P.S.); jaana.suvisaari@thl.fi (J.S.); katri.saaksjarvi@thl.fi (K.S.); tuija.jaaskelainen@thl.fi (T.J.); suvi.parikka@thl.fi (S.P.); seppo.koskinen@thl.fi (S.K.)

**Keywords:** COVID-19, people with disabilities, lifestyle, psychological distress, specific disability types

## Abstract

We investigated whether people with disabilities—cognition, vision, hearing, mobility, or at least one of these disabilities—report more COVID-19-related negative lifestyle changes than those without disabilities, and whether psychological distress (MHI-5) mediates the association between disabilities and negative lifestyle changes. Information about COVID-related lifestyle changes among people with disabilities is scarce. We analyzed population-based data from the 2020 FinSote survey carried out between September 2020 and February 2021 in Finland (*n* = 22,165, aged 20+). Logistic regressions were applied to investigate the effect of the COVID-19 pandemic and related restrictions on negative lifestyle changes—sleeping problems or nightmares, daily exercise, vegetable consumption, and snacking. To test for a mediation effect of psychological distress, the Karlson–Holm–Breen method was used. People with all disability types reported increased sleeping problems or nightmares, and decreased vegetable consumption during the pandemic more frequently than those without. People with mobility and cognitive disabilities more frequently reported decreased daily exercise. People with cognitive disabilities more often reported increased snacking. Psychological distress mediated associations between disabilities and negative lifestyle changes, with the highest association between cognitive disabilities and increased sleeping problems or nightmares (B = 0.60), and the lowest between mobility disabilities and decreased daily exercise (B = 0.08). The results suggest that strategies to promote healthy lifestyles should consider people with disabilities. Alleviating their psychological distress during crisis situations could be one approach.

## 1. Introduction

The COVID-19 pandemic has drastically changed everyday lives around the globe within a short period of time. To curb the spread of the virus, various containment measures, such as social distancing, working remotely, closure of leisure facilities, and restrictions on gatherings, have been instituted [1,2]. People with disabilities have been particularly vulnerable to many of the negative consequences of these measures [3,4,5,6,7,8,9,10,11,12,13].

Several clear lifestyle changes, e.g., sleep, daily exercise, and diet, have been reported in several countries during the pandemic [14,15,16,17,18,19,20,21,22,23,24]. Sleeping problems have increased, and daily exercise has decreased in the entire population [14,15,16,17,18,19], though some studies have found the opposite [20,21] or no changes [22,23]. Previous studies show increases or no changes in vegetable consumption during the COVID-19 pandemic [17,21,24,25]. Some systematic reviews have indicated that snacking increased during the pandemic among the general population [26].

Much less information is available about COVID-related lifestyle changes among people with disabilities. People with disabilities are a heterogenous group of individuals with varying levels of needs and abilities. Several subgroups can be identified, such as mobility, vision, hearing, and cognitive disabilities, and various disabilities can manifest together [7,13,27]. Disability severity can vary from mild to very severe, inducing different consequences for those with disabilities [28]. Therefore, grouping people with disabilities together gives a general picture of their lives, lifestyles, and experiences, which should be considered when interpreting the results.

Comorbidities are often more frequent among people with disabilities [27,29], and therefore, the impact of negative lifestyle changes can be particularly serious for this group. Steptoe et al. observed that, in the UK, older people (aged 50 years or more) with mobility disabilities reported more frequently that their sleep quality was impaired during the pandemic compared to those without disabilities [8]. In Ethiopia, sleeping problems have been highly prevalent during the pandemic among individuals with disabilities [6], and 66% of Norwegian people with disabilities reported decreased daily exercise in 2020 compared to the same period in 2019 [11]. However, in some of these studies, people with disabilities were not compared to people without disabilities, and different disability types were not investigated. Additionally, there is little information related to changes in eating habits among people with disabilities during the COVID-19 pandemic.

Thus, the scant evidence suggests more negative lifestyle changes among people with disabilities than among those without, but the reasons for this are not known. The role of psychological distress in producing or boosting unfavorable lifestyle changes deserves attention [15,30,31,32,33]. Higher psychological distress levels have been reported to associate with COVID-19-related negative lifestyle changes—increased sleeping problems [15,30], decreased physical activity [30,31], and increased unhealthy eating [32,33] among the general population. Furthermore, psychological distress and symptoms are particularly common among people with disabilities [34,35,36], and they have increased during the pandemic [5,6,7,8,9,12].

The distress can originate from various adverse and stressful circumstances that are more prevalent among people with disabilities even before the pandemic, e.g., poor health and activity limitations, loneliness, unemployment, financial insecurity, social exclusion and discrimination, and inadequate health and social services [34,35,37,38,39,40]. Many of these issues have further deteriorated during the pandemic, and people, particularly those with disabilities, have reported considerable concern about COVID-19 [3,5,7,8,9,10,13]. People with disabilities may also respond more strongly to the stress of the pandemic than those without disabilities because they may have underlying health conditions that heighten their risk of consequences [27], such as serious illness from COVID-19 and being hospitalized if symptomatic [41]. Some forms of disabilities might predispose individuals to distress more than others. For example, persons with cognitive disabilities have been shown to have difficulty handling stressful situations, and to need others’ help to cope with distress [42], increasing their vulnerability to distress. Psychological distress influences lifestyles through different mechanisms. Psychological distress may increase wakefulness, reduce sleep efficiency, and increase nightmares [43,44]. Psychological distress may also decrease the resources needed to commit to exercising or eating healthy, and individuals may exhibit increased emotional eating to cope with psychological distress [32,44,45]. Thus, higher levels of psychological distress among individuals with disabilities may contribute to the association between disabilities and negative lifestyle changes during the pandemic. However, to our knowledge, earlier studies have not addressed the role of psychological distress in mediating the connection between various types of disabilities and negative lifestyle changes during the pandemic.

This study was conducted in the Finnish context and has two aims. First, we investigated whether people with various disabilities—cognitive, vision, hearing, mobility, or at least one of these disabilities—reported COVID-19-related negative changes in sleeping problems or nightmares, snacking, vegetable consumption, and daily exercise more frequently than those without disabilities. Second, we studied direct and indirect (mediated by psychological stress) pathways between various types of disabilities and negative lifestyle changes during the pandemic. We hypothesized that part of the association between disabilities and unfavorable changes in lifestyle is mediated through psychological distress. This research may provide information to target support for people with disabilities to improve their lifestyles in crisis situations.

## 2. Materials and Methods

### 2.1. Data and Design

We analyzed population-based, cross-sectional data from the 2020 national survey on health, well-being, and service use in Finland (FinSote survey). The questionnaire was sent to a sample of 48,400 Finnish people aged 20 and older, and 22,165 (46%) of them participated. The survey was carried out between September 2020 and February 2021 by the Finnish Institute for Health and Welfare (THL) [46]. The survey was approved by the THL Ethics Committee. Weights were used in the analyses to restore the representativeness of the data. The calculation of weights was based on the inverse probability weighing method [47,48]. The weights were calculated using register-based information for the entire sample on age, sex, marital status, education, geographical area, and native language.

### 2.2. Measures

#### 2.2.1. Disability

The definition of disability was based on the Washington Group Short Set (WG-SS) of questions about vision, hearing, mobility, and cognition (remembering and learning) [49]. Disabilities were identified through two multi-part questions. The first question asked whether the participant could walk about half a kilometer without resting, read a normal newspaper text with or without glasses, and hear a conversation between several people with or without a hearing aid. The response categories were as follows: no difficulty, some difficulty, a lot of difficulty, and cannot do at all. The second question asked whether the respondent could memorize and learn new information. The response categories were very well, well, satisfactorily, poorly, and very poorly.

Mobility disabilities refer to those who reported having at least a lot of difficulties in walking. Vision disabilities refer to those who reported having at least a lot of difficulties in reading. Hearing disabilities refer to those who reported having at least a lot of difficulties in hearing. Cognitive disabilities refer to those who reported that they memorized and/or learned new information poorly or very poorly. In addition to variables of specific disability types, we also created a global variable indicating any disabilities: it was categorized as those with disabilities (having at least one specific disability) and those without disabilities (having none of the specific disabilities). We excluded 298 people from the analysis because of missing disability status.

#### 2.2.2. Outcome Variables

We used the following question about the COVID-19 pandemic’s effects on lifestyle: have the coronavirus pandemic or the subsequent restrictive measures affected your everyday life? The items were as follows: sleeping problems or nightmares, daily exercise, eating vegetables (including cooked, not potatoes), and snacking (eating sweets, chocolate, soft drinks, crisps, etc.). The response options were: no influence; yes, increased; yes, decreased; and not applicable. We report these negative lifestyle changes: increased sleeping problems or nightmares, increased snacking, decreased vegetable consumption, and decreased daily exercise. The question on snacking was presented only to those under 75 years of age.

#### 2.2.3. Potential Meditator

Psychological distress was identified based on the Mental Health Inventory (MHI)-5 [50]. This indicator is based on five questions: How much of the time in the previous 4 weeks: (a) Have you been a very nervous person? (b) Have you felt so down in the dumps that nothing could cheer you up? (c) Have you felt calm and peaceful? (d) Have you felt downhearted and blue? (e) Have you been a happy person? The response categories were: (1) all of the time, (2) most of the time, (3) a good bit of the time, (4) some of the time, (5) a little of the time, and (6) none of the time. The inner consistency for MHI-5 was good (Cronbach’s alpha 0.86). The scores in questions c and e were converted into reverse order, and the points were added (sum score between 5 and 30). The scale was converted to 0–100, with lower scores indicating higher psychological distress. People were defined as having clinically significant psychological distress if their score on the MHI-5 was 60 or below [51].

#### 2.2.4. Demographic Covariates

Age (20–54, 55–74, and ≥75 years), sex (female and male), living alone (yes and no), and level of education were selected as demographic covariates, because previous studies indicate that these variables are associated with lifestyles [6,14,16,18,24,52]. They were all significantly associated with at least one lifestyle change in our data (*p* < 0.05). The level of education was based on how many years in total the participant had attended school or studied full-time. To calculate the relative level of education (low, medium, high), the respondents were first divided by sex into 10-year age groups. Then, each age group was divided into three categories based on years of education, so that each class contained about one-third of the respondents.

#### 2.2.5. Data Analyses

All data analyses were conducted using Stata, version 16. We used the survey analysis procedures in Stata to analyze complex survey data by considering the sample design [53]. Frequency analyses were conducted to examine the prevalence of disabilities and the demographic data of people with and without disabilities. Adjusted logistic regression models were applied to compare each disability status (i.e., mobility, vision, hearing, cognitive, or any disabilities) to those without such disabilities in each outcome variable when the demographic covariates were controlled for. In these models, the adjusted prevalence of the outcome variables according to disability status was estimated using the margins command [54]. The same comparisons were examined when the demographic covariates were not controlled for in unadjusted logistic regression models. We report the odds ratio (OR) as a measure of association.

If the disability type was found to be significantly associated with the negative lifestyle change in the adjusted logistic regression model, mediation analyses were performed to identify the direct and indirect pathways (mediated through psychological distress) between disabilities and the lifestyle change (Figure 1). These analyses were adjusted for the demographic covariates.

For the mediation analysis, we used the Karlson–Holm–Breen (KHB) method, developed for nonlinear probability models, such as logit models [55,56,57], because the strategy described for linear models cannot be used in the context of logit models. A benefit of this method is that it allows the total effect of a variable to be decomposed into direct and indirect parts (mediation), and provides unbiased decompositions of the variable. Furthermore, it allows the inclusion of variables that control for confounding influences on decomposition. This method has been widely used in previous studies for mediation analysis in logistic regression [58,59].

We used the KHB method to decompose the total effect of the disability type on the negative lifestyle change in a logistic regression model into the sum of the direct and indirect mediation (through psychological distress) effects [55,56]. We reported the direct, indirect (mediation), and total (direct + indirect effect) unstandardized coefficients (B). Using the KHB method, we also estimated the percentage of psychological distress that accounted for the association between disabilities and lifestyle change (mediating percentage) by dividing the indirect mediation effect by the total effect. The KHB method was implemented by a user-written “khb” command in STATA; to adjust for the complex survey design, all the analyses were weighted [56].

## 3. Results

Approximately 13% of the adult population reported any disabilities (Table 1). Mobility (7%) and cognitive disabilities (6%) were the most common, whereas vision (2%) and hearing disabilities (3%) were less common. Table 1 presents the demographic characteristics across disability type. Those with disabilities were older, and their education level was lower compared to those without disabilities.

### 3.1. Lifestyle Differences between People with and without Disabilities

Table 2 presents adjusted ORs and the proportion of those reporting negative lifestyle changes among people with disabilities compared to those without disabilities. During the pandemic, people with any disabilities reported negative changes in all four lifestyle aspects more often than those without disabilities. The difference was largest in increased sleeping problems or nightmares (22% vs. 11%). People with cognitive disabilities differed from those without such disabilities in all lifestyle aspects, and increased snacking was found only among this group. Among the other types of disabilities, a few exceptions to the tendency of reporting more negative changes were noticed: those with vision and hearing disabilities did not differ from those without such disabilities in daily exercise and snacking. Furthermore, those with mobility disabilities did not report increased snacking more often than people without such disabilities. It is also noteworthy that persons with mobility disabilities had the highest odds for decreased exercise. The unadjusted results were quite similar to the adjusted results (Table A1).

### 3.2. Psychological Distress as a Mediator

We found higher levels of psychological distress among people with disabilities than among those without such disabilities (Table 3); the differences were highest between those with and without cognitive disabilities. Figure 1 presents the proposed mediation models, and Table 4 presents the results of the mediation models adjusted by the demographic covariates.

#### 3.2.1. Increased Sleeping Problems or Nightmares

Cognitive, vision, hearing, mobility, and any disabilities had an indirect association, via psychological distress, with increased sleeping problems or nightmares, but the direct pathway remained significant (Table 4). The results regarding cognitive disabilities are presented in Figure 2.

The results suggest that the association between each disability and increased sleeping problems or nightmares was mediated by psychological distress. Psychological distress accounted for 60% of the association between cognitive disabilities and sleeping problems or nightmares. The other mediating percentages varied between 42 to 49%.

#### 3.2.2. Decreased Daily Exercise

Any disabilities, as well as cognitive and mobility disabilities, also had an indirect association, via psychological distress, with daily exercise, but the direct association remained significant (Table 4). These results suggest that the association between these disabilities and decreased daily exercise was mediated by psychological distress. Psychological distress accounted for 35% of the association between cognitive disabilities and decreased daily exercise, but only 9% of the association between mobility disabilities and decreased daily exercise.

#### 3.2.3. Decreased Vegetable Consumption

Cognitive, hearing, mobility, and any disabilities had an indirect association, via psychological distress, with decreased vegetable consumption, but the direct association remained significant (Table 4). Vision disabilities had a significant indirect association, via psychological distress, with decreased vegetable consumption, whereas the direct effect became insignificant. This indicates that the association between these disabilities and decreased vegetable consumption was mediated by psychological distress. The contribution of psychological distress on the association between disabilities and decreased vegetable consumption was highest in cognitive and vision disabilities (45% and 40%), whereas the percentages were lower in other disabilities.

#### 3.2.4. Increased Snacking

Any and cognitive disabilities had an indirect association, via psychological distress, with increased snacking, but the direct association remained significant (Table 4). These results suggest that the association between disabilities and increased snacking was mediated by psychological distress. Psychological distress accounted for 42% of the association between cognitive disabilities and snacking.

## 4. Discussion

The large nationally representative data on the Finnish general adult population allowed us to disaggregate the results of COVID-19 effects on lifestyles by disabilities, and therefore, to provide new information about the situation of individuals with various disabilities during the pandemic in comparison to counterparts without disabilities. The main finding was that people with disabilities reported, more often than those without disabilities, negative COVID-19-related changes in lifestyles, such as sleeping problems or nightmares, snacking, vegetable consumption, and daily exercise. However, these changes varied somewhat across various disability types. The association between disabilities and negative lifestyle changes was mediated by psychological distress.

We found that people with various disabilities—cognitive, hearing, vision, mobility, and any disabilities—reported that the pandemic increased their sleeping problems or nightmares more often than those without disabilities. These results extend previous studies with similar findings among people with disabilities and older adults with mobility disabilities [6,8]. Our results suggest that COVID-19 may have deepened pre-existing differences in sleeping problems between people with various disabilities and those without disabilities [60,61,62,63]. Our study is the first to show that the association between various disabilities and increased sleeping problems or nightmares during the pandemic is mediated by psychological distress. We also found that, particularly, people with cognitive disabilities reported more psychological distress than those without disabilities, and their psychological distress accounted for 60% of the association between cognitive disabilities and sleeping problems or nightmares. Earlier studies have also indicated that people with disabilities experience higher psychological distress than those without [34,35,36]. People with cognitive disabilities may have difficulties understanding, dealing with, and finding solutions to the negative changes in their daily lives caused by the pandemic [64], thus decreasing their psychological well-being. As a consequence, their sleeping problems, such as wakefulness and reduced sleep efficiency, may increase [43].

We found that only people with cognitive and mobility disabilities reported that the pandemic decreased their daily exercise more often than those without disabilities; the difference was higher among those with mobility disabilities. Furthermore, those with vision and hearing disabilities did not differ from those without disabilities in terms of daily exercise. Our results extend previous evidence that people with disabilities perceived that their daily exercise decreased in 2020 [11]. The COVID-19 pandemic may have widened the pre-existing gap in daily exercise between people with cognitive and mobility disabilities and those without [29]. We further found that psychological distress mediated the association between cognitive disabilities and decreased daily exercise. During stressful times such as the pandemic, people are especially motivated to be physically active for their mental health, but may be too psychologically distressed to undertake exercise [45]. Higher psychological barriers among people with cognitive disabilities during the COVID-19 pandemic may thus have especially affected their daily exercise. Additionally, our results indicated that psychological distress accounted for only 9% of the association between mobility disabilities and daily exercise. Other issues potentially explaining the decreased daily exercise among people with mobility disabilities could include social distancing, the closure of regular exercise venues, and the need for more social support [45]. Those with mobility disabilities, who had the highest prevalence of daily inactivity before the pandemic [29], may have lower competence in daily exercise, and thus decreased their daily exercise under the restricted conditions. The closure of regular exercise venues can be serious for people with mobility disabilities because they may need these services to maintain their physical functioning. People with mobility disabilities, in particular, may need personal assistance and services (e.g., physiotherapy) in their daily exercise and transportation services to travel to exercise locations; however, the availability of such assistance and services declined during the pandemic [65].

We found that people with various disabilities more often reported that, during the pandemic, they decreased their vegetable consumption compared to those without disabilities, with the difference being highest among those with hearing disabilities. We also found that only those with cognitive disabilities more often reported that, during the pandemic, they had increased their snacking compared to those without. There is some evidence that, before the pandemic, people with cognitive disabilities ate less healthily than those without such disabilities [66]. Our findings showed that the pandemic could have widened this gap. Moreover, in this case, psychological distress has a significant role: it mediates the association between cognitive, vision, hearing, and mobility disabilities and decreased vegetable consumption. People with disabilities who are also psychologically distressed may have no resources to commit to eating healthy vegetables during the pandemic. In our results, the association between cognitive disabilities and snacking was also mediated by psychological distress. Particularly, people with cognitive disabilities may have difficulties regulating their emotions and controlling their behaviors [67]. Consequently, they may engage in emotional eating and snacking during the pandemic to deal with their negative emotions and stressors [32].

### Limitations and Strengths

The strength of this study was the large sample size representing the Finnish adult population. However, the response rate was relatively low, which weakened the generalizability of the results to the entire population. However, weights were used to correct for the bias caused by nonparticipation through the use of all the register data available for the entire sample. This improved the generalizability of the results.

Another strength is that we based our disability metrics on an internationally recognized and validated method—the WG-SS tool—to identify individuals with disabilities in general and those with specific types [49]. The minor differences between the questions used in this study and those in the WG-SS instrument hardly biased the inferences of this study. The national wordings were used to avoid breaking the existing time series. A clear shortcoming is that two of the domains in WG-SS, namely self-care and communication, were not included in our instrument, and therefore, may have affected the amount of people identified as disabled. However, the four domains (seeing, hearing, mobility, and cognition) used here have been considered as core domains, usable for general population surveys, where space restrictions are often an issue [68].

We also determined people’s psychological distress by applying the MHI-5, which is an internationally used instrument. Previous studies indicated that the constructive validity and inner consistency (also in our study) were acceptable for MHI-5 [50,69]. We found that psychological distress accounted for the association between disabilities and negative lifestyle changes during the COVID-19 pandemic. However, other causes can also lead to negative lifestyles changes [70]. Additionally, there are many individual varying reasons for psychological distress, such as poor health, activity limitations, loneliness, unemployment, financial insecurity, family–work conflict, social exclusion and discrimination, and inadequate health and social services [34,35,37,38,39,40,71]. Future research could further investigate the other components that may be related to associations between psychological distress and lifestyles during the COVID-19 pandemic.

One limitation of our study is that its cross-sectional nature did not allow us to address cause-and-effect relationships. However, we asked the participants directly about how COVID-19 had changed their lifestyles.

## 5. Conclusions

Our results showed that, compared to those without disabilities, people with various disabilities reported increased sleeping problems or nightmares, and decreased vegetable consumption during the pandemic. Furthermore, people with mobility disabilities, in particular, reported decreased daily exercise, and people with cognitive disabilities reported increased snacking. These negative changes can have serious consequences for public health among people with various disabilities, as this population has more chronic illnesses than non-disabled people [27,29]. Our results underline the need for targeted health promotion measures for people with various types of disabilities so that the identified negative lifestyle changes do not persist.

We further found that the association between disabilities and negative lifestyle changes, including increased sleeping problems or nightmares, snacking, decreased vegetable consumption, and daily exercise, was accounted for by psychological distress. Alleviating the psychological distress of people with various disabilities during crisis situations could, therefore, decrease these negative lifestyle changes. To address psychological distress among people with various disabilities, improved access to mental health screening and care are needed [7,36]. It is important to develop targeted mental health interventions and supportive means (e.g., social support, stress-coping strategies, and mindfulness) to ensure that psychological distress among people with disabilities does not become further entrenched [42,72,73]. The COVID-19 pandemic has brought an increased burden of psychological distress, especially for people with disabilities. Therefore, policymakers should ensure that there are sufficient resources in mental health services to relieve this burden.

## Figures and Tables

**Figure 1 ijerph-19-06971-f001:**
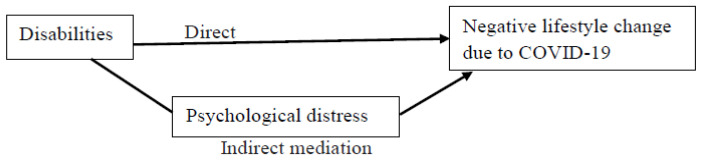
The proposed mediation model. Indirect mediation path = association between disabilities and negative lifestyle changes (i.e., increased sleep problems or nightmares, decreased daily exercise, decreased vegetable consumption, and increased snacking) through the mediation of psychological distress. Direct path = association between disabilities and negative lifestyle changes while controlling for the mediator.

**Figure 2 ijerph-19-06971-f002:**
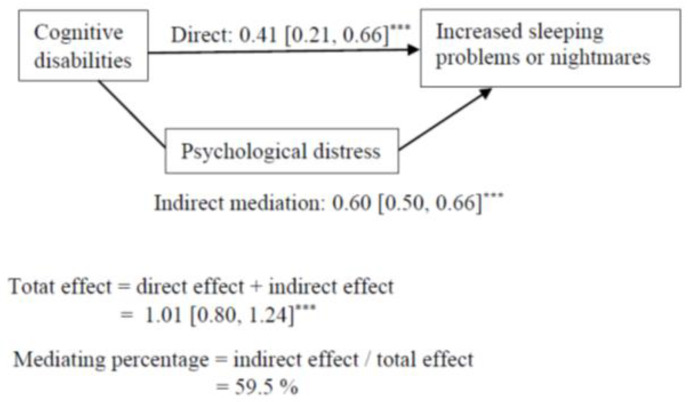
Mediation model for the association between cognitive disabilities and increasing sleeping problems or nightmares. Indirect mediation path = association between cognitive disabilities and increased sleeping problems or nightmares through the mediation of psychological distress. Direct path = association between cognitive disabilities and negative lifestyle changes while controlling for the mediator. Mediating percentage = the proportion of how much psychological distress accounted for the associations between cognitive disabilities and increased sleeping problems or nightmares. *** *p* < 0.001.

**Table 1 ijerph-19-06971-t001:** Prevalence of disabilities and sociodemographic characteristics by disability group, percentages, and confidence intervals (95% CI).

	No Disabilities	Any Disabilities ^a^	Cognitive Disabilities	Vision Disabilities	Hearing Disabilities	Mobility Disabilities
Total, *n*	17,733	4134	1773	663	884	2580
Total	86.8 [86.2, 87.3]	13.2 [12.7, 13.8]	6.3 [5.9, 6.8]	2.3 [2.1, 2.6]	2.7 [2.4, 2.9]	7.0 [6.6, 7.3]
Females	52.7 [51.6, 53.7]	51.4 [49.2, 53.7]	47.5 [44.0, 51.0]	46.5 [41.0, 52.1]	44.1 [39.5, 48.7]	58.0 [55.3, 60.7]
Age						
20–54	58.9 [58.0, 59.8]	27.0 [24.4, 29.6]	36.2 [32.2, 40.2]	34.1 [27.7, 40.5]	24.3 [18.8, 29.8]	12.0 [9.3, 14.7]
55–74	32.3 [31.4, 33.1]	35.3 [33.2, 37.4]	30.1 [27.1, 33.0]	31.5 [26.6, 36.4]	29.0 [25.0, 33.0]	36.3 [33.7, 38.9]
75 or over	8.9 [8.5, 9.2]	37.7 [35.8, 39.6]	33.7 [31.0, 36.4]	34.3 [29.9, 38.8]	46.6 [42.1, 51.2]	51.7 [49.0, 54.4]
Living alone	28.4 [27.4, 29.4]	45.8 [43.5, 48.0]	45.8 [42.2, 49.4]	47.5 [41.6, 53.4]	45.4 [40.5, 50.2]	49.4 [46.7, 52.1]
Education level						
Low	39.6 [38.6, 40.7]	54.1 [51.8, 56.5]	55.0 [51.3, 58.6]	65.4 [59.9, 70.9]	56.0 [51.1, 60.9]	53.1 [50.3, 55.8]
Medium	33.7 [32.8, 34.7]	29.3 [27.2, 31.4]	29.6 [26.3, 33.0]	24.4 [19.4, 29.4]	26.6 [22.4, 30.8]	30.1 [27.6, 32.6]
High	26.6 [25.7, 27.5]	16.5 [14.9, 18.1]	15.4 [12.9, 17.9]	10.2 [7.3, 13.1]	17.4 [13.8, 21.0]	16.8 [14.9, 18.7]

^a^ At least one disability.

**Table 2 ijerph-19-06971-t002:** Adjusted associations (OR) between disabilities and COVID-19-related negative changes in lifestyles, and the prevalence (%) of negative changes in lifestyles.

	**Increased Sleeping Problems or Nightmares**		**Decreased Daily Exercise**	
**Disabilities**	**OR** **[95% CI]**	**% [95% CI]**	**OR** **[95% CI]**	**% [95% CI]**
Any disabilities ^a^				
No	ref.	10.7 [10.0, 11.4]	ref.	24.0 [23.0, 24.9]
Yes	2.45 *** [2.06, 2.91]	22.4 [19.7, 25.0]	1.80 *** [1.57, 2.05]	35.8 [33.0, 38.5]
Cognitive disabilities				
No	ref.	11.3 [10.6, 12.0]	ref.	24.8 [23.9, 25.7]
Yes	2.73 *** [2.19, 3.40]	25.3 [21.4, 29.2]	1.56 *** [1.30, 1.88]	33.7 [29.8, 37.6]
Vision disabilities				
No	ref.	11.9 [11.2, 12.6]	ref.	25.1 [24.2, 26.1]
Yes	2.74 *** [1.95, 3.84]	26.4 [20.1, 32.6]	1.37 [0.99, 1.89]	31.2 [25.0, 37.5]
Hearing disabilities				
No	ref.	11.8 [11.2, 12.5]	ref.	25.2 [24.3, 26.1]
Yes	2.52 *** [1.85, 3.45]	24.8 [19.2, 30.4]	1.19 [0.90, 1.58]	28.6 [23.0, 34.1]
Mobility disabilities				
No	ref.	11.5 [10.8, 12.2]	ref.	24.2 [23.3, 25.1]
Yes	2.04 *** [1.68, 2.49]	20.8 [17.7, 23.9]	2.52 *** [2.18, 2.93]	43.9 [40.5, 47.4]
	**Decreased vegetable consumption**		**Increased snacking ^b^**	
**Disabilities**	**OR** **[95% CI]**	**% [95% CI]**	**OR** **[95% CI]**	**% [95% CI]**
Any disabilities				
No	ref.	4.6 [4.2, 5.1]	ref.	25.6 [24.5, 26.7]
Yes	2.05 *** [1.60, 2.63]	9.0 [7.2, 10.9]	1.71 *** [1.42, 2.05]	36.5 [32.6., 40.3]
Cognitive disabilities				
No	ref.	4.9 [4.4, 5.4]	ref.	25.9 [24.8, 26.9]
Yes	1.99 *** [1.45, 2.72]	9.2 [6.7, 11.7]	1.95 *** [1.52, 2.50]	39.8 [34.3, 45.4]
Vision disabilities				
No	ref.	5.1 [4.6, 5.5]	ref.	26.3 [25.3, 27.4]
Yes	1.80 * [1.12, 2.89]	8.7 [5.0, 12.4]	1.47 [0.95, 2.26]	34.0 [24.8, 43.3]
Hearing disabilities				
No	ref.	5.0 [4.5, 5.5]	ref.	26.3 [25.3, 27.4]
Yes	2.36 *** [1.60, 3.47]	10.9 [7.3, 14.6]	1.37 [0.86, 2.17]	32.6 [23.0, 42.2]
Mobility disabilities				
No	ref.	4.9 [4.4, 5.4]	ref.	26.2 [25.2, 27.3]
Yes	1.94 *** [1.48, 2.55]	9.1 [6.9, 11.2]	1.39 [1.00, 1.86]	33.1 [27.0, 39.2]

Adjusted for age, sex, living alone, and level of education; OR = odds ratio; CI = confidence interval; ref. = reference group; ^a^ At least one disability; ^b^ Restricted to persons under 75 years of age; * *p* < 0.05. *** *p* < 0.001.

**Table 3 ijerph-19-06971-t003:** Adjusted associations (OR) between disability and psychological distress, and the prevalence (%) of psychological distress.

	OR [95% CI]	% [95% CI]
Any disabilities ^a^		
No	ref.	18.6 [17.7, 19.4]
Yes	4.98 *** [4.36, 5.68]	51.1 [48.4, 53.9]
Cognitive disabilities		
No	ref.	20.0 [19.1, 20.8]
Yes	6.46 *** [5.46, 7.63]	59.9 [56.2, 63.5]
Vision disabilities		
No	ref.	21.7 [20.8, 22.5]
Yes	3.85 *** [2.96, 4.99]	50.3 [44.1, 56.5]
Hearing disabilities		
No	ref.	21.5 [20.7, 22.4]
Yes	4.54 *** [3.67, 5.62]	54.0 [49.0, 59.0]
Mobility disabilities		
No	ref.	20.9 [20.1, 21.8]
Yes	3.14 *** [2.70, 3.65]	44.2 [40.8, 47.6]

Adjusted for age, sex, living alone, and level of education; ^a^ At least one disability; *** *p* < 0.001.

**Table 4 ijerph-19-06971-t004:** Psychological distress mediates the associations between disabilities and negative changes in lifestyles.

	Any Disabilities ^a^	Cognitive Disabilities	Vision Disabilities	Hearing Disabilities	Mobility Disabilities
Mediation Models	B [95% CI]	B [95% CI]	B [95% CI]	B [95% CI]	B [95% CI]
Disabilities → Increased sleeping problems or nightmares					
Direct path	0.44 *** [0.26, 0.62]	0.41 *** [0.21, 0.66]	0.64 *** [0.28, 1.00]	0.54 ** [0.21, 0.87]	0.43 *** [0.22, 0.64]
Indirect path through MHI-5	0.44 *** [0.38, 0.51]	0.60 *** [0.50, 0.66]	0.44 *** [0.33, 0.55]	0.48 *** [0.39, 0.57]	0.31 *** [0.25, 0.37]
Total	0.89 *** [0.71, 1.06]	1.01 *** [80, 1.24]	1.08 *** [0.73, 1.44]	1.02 ** [0.69, 1.35]	0.74 *** [0.53, 0.95]
Mediating percentage ^b^, %	49.2	59.5	40.8	47.0	41.7
Disabilities → Decreased daily exercise			ns	ns	
Direct path	0.48 *** [0.34, 0.62]	0.31 ** [0.11, 0.51]			0.85 *** [0.70, 1.01]
Indirect path through MHI-5	0.11 *** [0.08, 0.15]	0.17 *** [0.11, 0.22]			0.08 *** [0.06, 0.11]
Total	0.60 *** [0.46, 0.73]	0.48 *** [0.28, 0.67]			0.94 *** [0.79, 1.09]
Mediating percentage, %	19.2	34.8			9.0
Disabilities → Decreased vegetable consumption					
Direct path	0.45 *** [0.18, 0.72]	0.37 * [0.03, 0.72]	0.34 [−0.14, 0.83]	0.66 *** [0.24, 1.08]	0.43 *** [0.14, 0.71]
Indirect path through MHI-5	0.22 *** [0.15, 0.30]	0.30 *** [0.20, 0.40]	0.24 *** [0.15, 0.32]	0.24 *** [0.16, 0.32]	0.16 *** [0.11, 0.22]
Total	0.67 *** [0.42, 0.93]	0.68 *** [0.35, 1.00]	0.58 * [0.10, 1.07]	0.90 *** [0.49, 1.30]	0.59 *** [0.30, 0.87]
Mediating percentage, %	32.7	44.7	39.9	26.4	27.6
Disabilities → Increased snacking ^c^			ns	ns	ns
Direct path	0.30 ** [0.11, 0.50]	0.38 ** [0.12, 0.65]			
Indirect mediation path through MHI-5	0.22 *** [0.16, 0.27]	0.28 *** [0.21, 0.35]			
Total	0.52 *** [0.33, 0.71]	0.66 *** [0.40, 0.92]			
Mediating percentage, %	41.5	42.0			

Adjusted for age, sex, living alone, and level of education; MHI-5 = psychological distress (the Mental Health Inventory-5); B = unstandardized effect estimates; CI = confidence interval; ns = no significant association between disabilities and negative lifestyle change; ^a^ At least one disability; ^b^ The proportion of how much MHI-5 accounted for the associations between disabilities and lifestyle changes; ^c^ Restricted to persons under 75 years of age. * *p* < 0.05. ** *p* < 0.01. *** *p* < 0.001.

## Data Availability

The data that support the findings of this study are not publicly available. The data from which personal data have been eliminated may be disclosed for research purposes from the Social and Health Data Permit Authority (Findata) in return for a research proposal and an approved user authorization application.

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
