# Peer review of "Differences in Unfavorable Lifestyle Changes during the COVID-19 Pandemic between People with and without Disabilities in Finland: Psychological Distress as a Mediator"

_ijerph, 2022, doi:10.3390/ijerph19126971_

Round 1

Reviewer 1 Report

The topic of the study is interesting. The research methodology is correct and the statistical methodology is appropriate in relation to the main objective of the study. The results are well described and the multivariate analyses allow for the removal of possible confounders. The descriptive tables of the results are clear. The strengths and weaknesses of the study are also described. The study can be published. 

Reviewer 2 Report

The study idea is smart, the sample size is amazingly high, and to my understanding, data are novel.

Abstract; this is well written, though one might question as to why vegetable consumption was an important factor, compared for instance to protein intake or to regular exercising. “…

 but the strength of the mediation effect …” here, the reader might want specifications how strong such strengths were. Given that the COVID-19 pandemic and its possible restrictions lasted for about 24 months, the authors might specify the more precise time point.

Introduction: As regards the zero-impacts or even favorable impact of the COVID-19 pandemic on sleep and physical activity, please see (Albrecht et al., 2022) and (Aghababa et al., 2021). In these two papers, the authors of the present manuscript will find further helpful references for the 0-association between the COVID-19 pandemic and its social consequences on physical activity and sleep. Overall, the Introduction section was well written.

Materials and Methods; well written.

Statistics: the authors applied the so-called Holm–Breen (KHB) method; given that this approach is virtually unknown to the well-established procedure of Kenny and Baron (1986), the reader needs a justification. I do not say that the KHB is wrong! I just say, that the procedure is unknown and that, accordingly, the reader needs more help to understand the underlying rationale. Further, if the KHB suits well for non-linear probability models, does is mean that the dimensions were basically associated in a non-linear fashion?

Results: The reader would benefit from Figure 1, though with the appropriate and specific statistical indices. In doing so, this would help to understand the relative importance (explained variances) of the direct and indirect effects.

Discussion and Conclusions: very well written.

References

Aghababa, A., Zamani Sani, S. H., Rohani, H., Nabilpour, M., Badicu, G., Fathirezaie, Z., & Brand, S. (2021). No Evidence of Systematic Change of Physical Activity Patterns Before and During the Covid-19 Pandemic and Related Mood States Among Iranian Adults Attending Team Sports Activities. Front Psychol, 12, 641895. doi:10.3389/fpsyg.2021.641895

Albrecht, J. N., Werner, H., Rieger, N., Widmer, N., Janisch, D., Huber, R., & Jenni, O. G. (2022). Association Between Homeschooling and Adolescent Sleep Duration and Health During COVID-19 Pandemic High School Closures. JAMA Netw Open, 5(1), e2142100-e2142100. doi:10.1001/jamanetworkopen.2021.42100

Baron, R. M., & Kenny, D. A. (1986). The moderator–mediator variable distinction in social psychological research: Conceptual, strategic, and statistical considerations. J Pers Soc Psychol, 51(6), 1173-1182. doi:10.1037/0022-3514.51.6.1173

Reviewer 3 Report

Dear Authors,

While appreciating your detailed research I will suggest some revisions to make it come closer to a social approach towards respondents' lives.

I Title, Abstract and Introduction

All these sections need to include information that the respondents are from Finland. The mention about FinSoteSurvey does not convey this.

II Research Methods, Results and Discussion

Although it is easy to understand that internationally used and even validated variables are used to secure comparison with earlier results, this choice makes the research somewhat bleak, especially the lifestyle variables. Human choices about their modes of living, guided by values and attitudes, seem to be reduced to psychological habit and problems that are easily taken to go through statistical measures. Concerning the potential mediator variable, experienced and expressed distress is left unexplained, as a general phenomenon, without individually varying causes. Only the lifestyle changes are understood as due to the pandemic, but I see that a range of causes and the reason for taking them into account should be at least mentioned. 

The variables as such are clearly described and tables are presented carefully, reflecting the analysis done.

III Conclusions

The paper needs proper examples of practical implications: how to support disabled people to cope with distress in their daily life; who/which type of social or health organizations should be doing that and by what steps.

All the best for your revisions

Reviewer

Reviewer 4 Report

The paper addresses an important topic in health-behavior research
during the COVID-19 pandemic. The results are interesting.
Further strengths are the huge sample size and the analytical approach.

The conceptual frameworks integrating the results found could be more
elaborated.
Are people with disabilities simply more vulnerable, or why they seem to
have more difficulties to psychologically adapt to stressful events, and
through which mechanisms is this influencing their health behavior?

The practical implications could be discussed with more detailed
examples from everyday life.
How specific interventions / social policies could look like in detail?

Throughout the paper, the authors often speak about disabled people in
general, but this seems a heterogeneous group, with special needs and
capacities for each individual.
An individual perspective seems more appropriate.
At least in terms of writing, this could be more attentively considered.

Round 2

Reviewer 4 Report

Differences in unfavorable lifestyle changes during the COVID-19 pandemic between people with and without disabilities: Psychological distress as a mediator

All requests were addressed throughout the paper. I don't have further comments on the paper.